# Astrocytic Factors Controlling Synaptogenesis: A Team Play

**DOI:** 10.3390/cells9102173

**Published:** 2020-09-26

**Authors:** Giuliana Fossati, Michela Matteoli, Elisabetta Menna

**Affiliations:** 1Humanitas Clinical and Research Center—IRCCS—NeuroCenter, via Manzoni 56, 20089 Rozzano, Milan, Italy; Giuliana.Fossati@humanitasresearch.it (G.F.); michela.matteoli@hunimed.eu (M.M.); 2CNR, Department of Biomedical Sciences, Institute of Neuroscience—URT Humanitas, via Manzoni 56, 20089 Rozzano, Italy

**Keywords:** astrocyte factors, astrocyte diversity, synaptogenesis, synaptic pruning, synaptic plasticity

## Abstract

Astrocytes are essential players in brain circuit development and homeostasis, controlling many aspects of synapse formation, function, plasticity and elimination both during development and adulthood. Accordingly, alterations in astrocyte morphogenesis and physiology may severely affect proper brain development, causing neurological or neuropsychiatric conditions. Recent findings revealed a huge astrocyte heterogeneity among different brain areas, which is likely at the foundation of the different synaptogenic potential of these cells in selected brain regions. This review highlights recent findings on novel mechanisms that regulate astrocyte-mediated synaptogenesis during development, and the control of synapse number in the critical period or upon synaptic plasticity.

## 1. Introduction/Overview

Astrocytes constitute the most abundant glial cells in the human brain [1]. A single cortical astrocyte enwraps multiple neuronal cell bodies and dendrites. It contacts more than 100,000 synapses through its finer perisynaptic astroglial processes (PAPs) [2,3].

Through these contacts, astrocytes monitor and modulate synaptic function, thus actively controlling synaptic transmission. The close structural and functional interaction of the perisynaptic astrocytic processes with the neuronal pre- and postsynaptic structures led to the “tripartite synapse” concept [4], with astrocytes modulating synaptic transmission via the release of gliotransmitters, such as glutamate, D-serine and ATP, and neurotransmitter reuptake from the active zone. The “tripartite synapse” concept introduced to the scientific community the key role of astrocytes in information transfer and storage in the adult brain (see reviews [5,6,7,8]).

However, the relation between astrocytes and neurons starts already during brain development, when astrocytes play key roles in neuronal circuit formation, in particular controlling synapse assembly and maturation, as well as synaptic refinement. In this review, we will provide an updated summary of the astrocyte-secreted molecules involved in synaptogenesis, and we will focus on the recently discovered mechanisms that regulate astrocyte-mediated synaptogenesis. We will also present an overview of how astrocytes exert a tight control over synapse number, which goes well beyond the synaptogenic period, affecting the structure and function of the mature synapse.

Accumulating evidence indicates that alterations to astrocyte morphogenesis and physiology in this period may severely affect proper brain development, causing neurological or neuropsychiatric conditions. In particular by using patient-derived astrocytes, recent studies unveiled the potential involvement of these cells in the pathophysiology of neurodevelopmental disorders, such as Rett and Down syndromes. Neurons cultured with astrocytes differentiated from Rett syndrome patient-specific induced Pluripotent Stem Cells (iPSCs) show synaptic defects [9], whereas astrocytes differentiated from Down syndrome patient-specific iPSCs display altered transcriptomic profiles, altered adhesion capability and increased cellular dynamics, and may impact on neuronal synaptogenesis [10,11]. Not only genetic defects but also the lengthy use of general anesthetics in infants has been reported to affect astrocyte morphogenesis and be associated with neurobehavioral deficits [12]. In this review, we also reported the evidence that links specific astrocyte factors, involved in synaptogenesis or in the regulation of synaptic function, to neurological and neuropsychiatric conditions.

## 2. Astrocyte-Secreted Factors Induce Synaptogenesis

The synaptogenic role of astrocytes was initially discovered using a purified retinal ganglion cell (RGC) culture system. RGC neurons grown in the absence of astroglia form very few synapses. However, synapse formation is increased upon the addition of astrocyte-conditioned media (ACM) [13]. Since then, a large number of studies demonstrated that astrocytes are essential players in promoting synaptogenesis, particularly during brain development, and provided evidence for many astrocyte-secreted factors, including proteins, lipids and small molecules that control different aspects of excitatory and inhibitory synapse formation and maturation (Table 1).

During the last two decades the astrocyte “secretome”, i.e., the entire set of secreted proteins according to the definition coined by Tjalsma et al., in 2000 [42,43], has been extensively analyzed in the search for synapse-promoting factors. The full list of astrocytes’ molecules involved in synaptogenesis is reported in Table 1, and has been extensively reviewed in the last few years (see also [44,45,46,47]). Here we focus on three classes of well-known synaptogenic factors, which play key roles in early synaptogenesis through recently discovered molecular mechanisms.

### 2.1. Thrombospondins and Pentraxin 3

In the first two weeks of postnatal mouse development, which correspond to a period of massive synaptogenesis in most brain areas, astrocyte-derived thrombospondin 1 and 2 (TSP1, TSP2) are highly expressed and initiate the formation of synapses [14]. This process occurs through TSP binding membrane proteins, including the Ca^2+^ channel subunit α2δ-1 [15] or the postsynaptic cell adhesion molecule Neuroligin1 [48]. It has been demonstrated that TSPs induce the formation of ultrastructurally normal synaptic contacts, which are presynaptically active with cycling synaptic vesicles. Remarkably, they become postsynaptically silent because of the lack of functional 2-amino-3-(5-methyl-3-oxo-1,2-oxazol-4-yl)propanoic acid (AMPA) receptors [14].

An elegant study by Risher and colleagues recently demonstrated that in the cerebral cortex, TSP–α2δ-1 interaction controls synaptogenesis by acting postsynaptically via Rac1 [16]. Indeed, they found that the synaptogenic function of α2δ-1 is cell autonomous to neurons, requiring the postsynaptic expression of α2δ-1. Upon TSP–α2δ1 interaction at the postsinaptic level, the guanine nucleotide exchange factors (GEFs) Kalirin-7 and β-Pix/Cool-1 β activate the Rac1 pathway and promote the remodeling of actin cytoskeleton at the nascent synaptic contact [16] (Figure 1). TSP1 and TSP2 display a developmentally regulated pattern of expression and are downregulated around the second and third weeks of postnatal development [14]. Concomitantly, the glial secretion of TSP4 increases [49]. TSP4 differs from TSP1 and TSP2 for the lack of the procollagen and properdin-like (type I) repeats that enable interactions with glycosaminoglycans and integrin-associated proteins, and promote the activation of TGFβ [50]. Furthermore, whereas TSP1-deficient and TSP2-deficient mice display a significant decrease in the frequency of excitatory synapses [14,51], TSP4-deficient mice show increased synaptic activity due to the lack of regulation of presynaptic Ca^2+^ channels [52]. Thus, the developmental shift of TSPs subtypes’ activity during brain development may affect different aspects of synaptogenesis that need to be further explored. It has been known for a long time that the activity of TSPs is fundamental in promoting the formation of early synapses, which are silent. Which is/are the factors involved in the activation of these first synapses was not known until very recently. Indeed, it has been recently demonstrated that, concomitantly to TSP secretion, astrocytes release the innate immune molecule pentraxin3 (PTX3), which promotes the functional maturation of excitatory synapses formed during the first wave of synaptogenesis by inducing AMPA receptors’ clustering at the synapse [29].

This occurs through a process involving the key PTX3 binding partner, TSG6, the remodeling of the perineural extracellular matrix surrounding synaptic contacts, and requires β1-integrins (Figure 1). Quite unexpectedly, PTX3-deficient mice display weaker excitatory synapses in the hippocampus, not only at young ages but also at P30, indicating that a lack of endogenous PTX3 results in synapse defects, which apparently cannot be rescued by other astrocyte-derived factors expressed at later developmental stages [17,30,31] (see next paragraphs). Notably, PTX3 is able to interact with TSP1 and 2, but not with TSP4. Upon binding to TSP1, PTX3 activity is inhibited, thus representing an additional mechanism of control of the process [29]. Notably, PTX3 and TSP1 display a spatially and temporally overlapped expression also in the human brain, being higher in the astrocytes of the fetal cerebral cortex [53]. The relative amount of the two molecules could therefore be crucial to set the proper balance between synaptic growth and synapse maturation during the period of early synaptogenesis. Both common and rare variants of the gene encoding for TSP1, *THBS1*, have been found to be associated with Autism Spectrum Disorders (ASD) in a cohort of 313 patients by Sanger sequencing [54]. Furthermore, single nucleotide polymorphisms (SNPs) of the *PTX3* gene have been reported to result in PTX3 deficiency in humans, and are associated with a changed susceptibility to infections and an altered inflammatory response [55,56,57]. Whether these PTX3 variants might be linked to altered synaptogenesis under physiological conditions or in response to an inflammatory insult occurring during brain development is presently not known.

In addition to PTX3, another glial factor which increases AMPA receptors (AMPAR) levels in synapses has been recently identified in the astrocyte secretome Chordin-like 1 (Chrdl1), whose expression in the cerebral cortex peaks at P12–P14, a bit later than PTX3. In particular, Chrdl1 has been shown to stimulate the insertion of GluA2-containing calcium impermeable AMPARs at the synapse, thus regulating the so-called AMPAR switch underlying the functional maturation of excitatory synapses in the cerebral cortex [32].

### 2.2. Hevin and SPARC

Besides TSPs, two members of the secreted protein acidic and rich in cysteine (SPARC) family proteins, Hevin (also known as SPARC-like1/SPARCL1) and SPARC, were recognized as astrocyte-secreted factors that control synapse formation between cultured RGCs (Figure 2). Similar to TSPs, the treatment of RGCs with purified Hevin is sufficient to induce the formation of structurally mature but postsynaptically silent synapses [17]. Analyses of developing visual cortices in Hevin-null mice revealed fewer thalamocortical synapses and spines displaying features of immaturity, indicating that the protein is required for the proper development of thalamocortical synaptic connectivity [18]. Astrocyte-derived Hevin has been demonstrated to serve as a bridge for the non-interacting isoforms of neurexin-1alpha and neuroligin-1B at the level of thalamocortical connections in the developing visual cortex [19]. By contrast, SPARC antagonizes the synaptogenic action of Hevin through a dominant-negative action, most likely by interfering with the ability of Hevin to bridge neurexin-1alpha and neuroligin-1B [17]. In addition, SPARC is able to trigger a cell-autonomous program of synapse elimination [58]. In the developing mouse brain, Hevin and SPARC are expressed at high levels in astrocytes during the second and third weeks, which coincide with periods of synapse stabilization and synaptic refinement. In the adult brain, SPARC is downregulated, whereas Hevin expression remains high [17], in line with Hevin’s other role in synaptic maintenance and repair [59], as well as in synaptic plasticity (see below, chapter “Astrocyte-derived molecules affect synaptic plasticity”). Of note, *SPARCL1* has been recently identified as part of a network of genes linked to neuronal damage in the preclinical stages of Alzheimer’s disease (AD) [60]. In particular, *SPARCL1* variants that correlate with lower gene expression levels in the brain are associated with accelerated cognitive decline during preclinical AD.

### 2.3. Glypicans and Neuronal Pentraxins

Recently, astrocyte-secreted glypicans 4 and 6 have been discovered by using the same experimental setting (i.e., RGC cultures and ACM) that led to the discovery of thrombospondins. Glypicans are expressed at later stages of development (second and third postnatal weeks). The exposure of RGC cultures to glypicans 4 or 6 resulted in enhanced formation of active excitatory synapses containing GluA1 receptors, in opposition to the action of TSPs [30]. GPC4-deficient mice show defective excitatory synapses in the hippocampus [30] and alterations in social behavior, which resemble ASD core symptoms and that can be linked with loss of GluA1 [61]. The ability of glypican 4 to induce active synapses involves neuronal pentraxin 1 (NP1), which is released from presynaptic terminals by signaling through presynaptic protein tyrosine phosphatase receptor delta (PTPRδ). NP1 then stimulates AMPA receptors’ clustering on the postsynaptic dendrite, thus allowing the assembly of functional synapses [31] (Figure 3).

Neuronal pentraxins—the membrane-bound “neuronal pentraxin receptor” (NPR) and the secreted proteins NP1 and NARP (i.e., NP2)—are expressed by neurons and promote AMPA clustering at excitatory synapses by direct binding of the N- terminal domain of the receptor [62,63,64,65,66,67,68]. Recently, it has been demonstrated that the knockdown of NPR in hippocampal neurons dramatically decreases the assembly and function of both excitatory and inhibitory postsynaptic specializations [69]. Evidence showed that NPR recruits and stabilizes NP1 and NARP on the presynaptic plasma membrane, suggesting that neuronal pentraxins act as trans-synaptic organizers of both excitatory and inhibitory synapses. Diseases associated with GPC4 include Simpson-Golabi-Behmel Syndrome, which in some cases results in mild to severe intellectual disability [70].

## 3. Astrocyte Diversity Affects Their Own Synaptogenic Potential

Accumulating evidence indicates that the synaptogenic properties of astrocytes from diverse brain regions are different, indicating the specialization of local astrocyte populations within specific synaptic circuits. In line with specific astrocyte secretome varying according to the brain region, it has been reported that astrocytes from the cerebellum induce more synapses with respect to astrocytes from the midbrain, hippocampus or cortex, due to the higher expression of Hevin and glypican 4 in this population [71].

While astrocyte diversity has remained for a long time poorly investigated because of technical issues hampering their isolation from specific brain regions, recently developed techniques for single cell RNA profiling and proteomic studies largely advanced this line of investigation [49,72,73]. For instance, the fusion of reporter tags to ribosomal subunits allowed the isolation and analysis of actively translated mRNAs from astrocytes [72,74,75], while the use of magnetic-activated cell sorting preserved astrocyte morphology better than FACS [76,77]. Taking advantage of this technical improvement, Batiuk and colleagues recently identified, in the adult brain, molecularly and functionally diverse subpopulations of astrocytes, which eventually exert different roles in supporting synaptogenesis [78]. In particular, they recognized, throughout the cortex, astrocyte subtypes 2 and 3 (AST2 and 3), which differentially express genes involved in neurotransmission. AST2 is enriched in transcripts linked to glutamatergic neurotransmission, while AST3 is enriched in transcripts associated with GABAergic neurotransmission, thus suggesting that neuronal-derived signals induce astrocyte diversity at the local level [78].

Although deciphering the mechanisms responsible for the acquisition of astrocyte molecular and functional diversity is at its infancy, with intrinsic astrocytic factors and extrinsic neuronal signals being likely involved [79], it is becoming evident that astrocyte specialization—in space and time—dynamically regulates the synaptogenic potential of these cells. This mainly occurs through modifications of the astrocyte secretome composition throughout the lifespan. Of note, the synaptogenic effect of astrocytes was found to rely on the prevalence of a specific astrocyte population subtype, called population ‘C’, that is characterized by the presence of surface markers CD51, CD63 and CD71 [80]. This astrocyte subpopulation, isolated from an ALDH1L1–eGFP adult mouse model, is highly enriched for genes associated with synapse assembly and function. Consistently, when co-cultured with cortical neurons, C astrocytes are able to increase the frequency of both cortical excitatory and inhibitory currents [80].

## 4. Astrocytes Control Synaptic Pruning

Neurons generate an excess of synapses during development, and some of them are subsequently eliminated to reach a precise neuronal circuit assembly through a process called “pruning” [46,81]. This synaptic refinement happens in a precise time window termed the “developmental critical period”, a brief interval wherein neural circuits can be modified by sensory inputs. Within critical periods, the unnecessary and weak synapses are eliminated, whereas the remaining inputs are further strengthened to form the mature neural circuit. Afterwards, synapse elimination continues in the mature nervous system, through experience-dependent structural synaptic plasticity, even if the number of elimination events drops with age [82].

Traditionally, microglia have been thought as the only glial cells responsible for synaptic pruning [83], but accumulating evidence showed that astrocytes also exert a key role in regulating synaptic elimination during development [34,45,84]. The phagocytic ability of astrocytes has been demonstrated previously both in vitro, by using cell cultures, and in vivo in models of injury [44]. For example, in glaucoma, astrocytes have been demonstrated to constitutively perform phagocytosis through internalizations of large portions of axonal cytoplasm and axon-derived organelles [85]. Interestingly, similarly to microglia [86,87], the phagocytic activity of astrocytes appears to be influenced by sex, as recently demonstrated by the use of synthetic steroids which induce a significantly higher phagocytic activity in astrocytes derived from female mice under resting and inflammatory conditions [88].

The astrocyte-mediated phagocytosis of synapses, which is driven by network activity and is crucial in shaping the circuit in response to experience, may involve direct or indirect mechanisms. The direct phagocytosis of supernumerary synapses in the developing brain occurs via the astrocyte phagocytic receptors MERTK and MEGF10 [89]. Both these pathways are activated by an “eat-me” signal, like phosphatidylserine exposure, as occurs in the case of microglia [90]. MEGF10 (Multiple EGFLike Domains 10) is a known mediator of phagocytosis. Its action involves different intracellular proteins, but the exact function of this protein is still unknown [82]. MERTK (Mer Proto-oncogene Tyrosine Kinase) exploits the integrin pathway to control the actin cytoskeleton during phagocytosis. These pathways have been found to be active in the retinogeniculate system, and indeed MEGF10- and MERTK-deficient mice display defective synaptic pruning and, therefore, impaired eye specific segregation in the dorsolateral geniculate nucleus (dLGN) [46,82,89].

Astrocytes can also promote indirect phagocytosis by mediating the neuronal expression of phagocytic markers recognized by microglia. In this respect, the pioneer study by Bialas and Stevens showed that astrocytes indirectly regulate synapse elimination via the secretion of the transforming growth factor-β (TGF-β) in the retina. Astrocytes release TGFβ that stimulates C1q expression in RGC neurons. C1q is the initiating protein of the classical complement cascade that starts the complement-dependent synaptic refinement in the retinogeniculate system mediated by microglia. Accordingly, the inhibition of TGFβ resulted in defects in eye-specific segregation in the dLGN resembling microglia-deficient phenotypes [91,92].

In the last few years, direct evidence linking synapse elimination with human diseases has been emerging. Using reprogrammed in vitro model of microglia-mediated synapse engulfment, increased synapse elimination in patient-derived neural cultures and isolated synaptosomes was demonstrated [93]. This evidence added to more indirect indications supporting a link between defective synapse refinement and human diseases. Indeed, altered synaptic pruning has been reported in mice models with mutations of PTEN, which accounts for ~10% of cases of ASD [94]. It has been demonstrated that the genetic lack of the microglia receptor TREM2 impairs the ability of microglia to eliminate synapses and results in ASD-like phenotype, and that an impairment in synaptic material degradation in atg7-deficient mice microglia causes social behavioral defects and repetitive behaviors typical of ASDs [95].

Consistently, alterations in synapse number and structure and abnormalities in glial cells functioning have been found in many neurological disorders [96,97,98,99,100]. Given the central role of astrocytes, together with microglia, in synapse elimination, and based on the tight cooperation between astrocytes and neurons in brain development and function, it is reasonable to hypothesize that alterations in astrocyte-dependent synaptic pruning could be a concurrent factor in brain pathologies. Despite the difficulty in studying human pathologies and the paucity of tools to investigate specifically astrocytes hampering this line of investigation in the past, important contributions to this field are expected in the future.

Another new piece of evidence for the indirect mechanism of phagocytosis involves the proinflammatory cytokine Interleukin33 (IL33), which is expressed specifically by astrocytes in the CNS during the critical period of synaptic refinement. Vainchtein and colleagues demonstrated that IL33 directly mediates the developmentally regulated removal of excitatory connections between the reticular and the ventrobasal nuclei of the thalamus, increasing microglial phagocytic ability. Indeed, astrocyte-IL33 expression increases following synaptic maturation, suggesting a homeostatic loop by which astrocytes react to increased synapse number, promoting the process of synaptic pruning operated by microglia. On the contrary, IL33 deficiency has been associated with an excessive number of excitatory synapses and overall hyperexcitable intrathalamic circuit compared to wild-type controls [101,102,103]. The mechanism by which astrocyte-derived IL33 regulates microglial synaptic function is a new function for this cytokine, best known as an “alarmin” involved in tissue homeostasis. It is released by dying cells, and is crucially involved in wound healing and in the inflammatory response after injury [38]. IL33 also retains a role of “alarmin” in the adult CNS; in fact, in a mouse model of spinal cord injury, IL33 was upregulated by spinal cord astrocytes and promoted the resolution of inflammation through a mechanism involving TNFα [104].

## 5. Astrocyte-Derived Molecules Affect Synaptic Plasticity

### 5.1. Role of Astrocytes in Critical Period

Some glial factors with a recognized synaptogenic role at early stages of development are involved also in the subsequent phase of synapse refinement during the critical period. For example, Hevin expression remains high during the critical period [17], and it is required for the proper establishment of thalamocortical synapses. Hevin-deficient mice display an increased number of immature spines and the altered localization of excitatory synapses, indicating that a lack of Hevin results in abnormal thalamocortical connectivity probably due to the defective synaptic refinement process [18]. Further, Hevin acts as a bridge between the neuronal neurexin 1α and neuroligin 1B, which do not interact directly. This interaction recruits NMDA receptors at the synapse. Mice lacking Hevin display impaired ocular dominance plasticity, an event wherein synapses in the visual cortex remodel in response to changed visual experience. The rescue of Hevin expression specifically in astrocytes of the visual cortex before the closure of the critical period restores ocular dominance plasticity, showing that the expression of Hevin in astrocytes is sufficient to control this form of plasticity [19]. Furthermore, neuroligins are also expressed by astrocytes, and through their neuronal partner neurexin 1 control astrocytic morphogenesis and excitatory synaptogenesis [34]. A recent finding proposes astrocytic neuroligins for ensuring proper critical period termination in the developing Drosophila motor circuit. Indeed, in mutant drosophila, reduced neuroligin–neurexin signaling results in dendritic microtubules destabilization, enhanced dendrite dynamics, and impaired locomotor behavior [105].

Chordin-like 1 (Chrdl1) is another example of an astrocyte-secreted protein which is expressed throughout postnatal development and in the critical period. Besides being involved in the maturation of excitatory AMPAR-containing synapses during synaptogenesis, it plays a key role in regulating the closure of the critical period in the maturing brain. In the absence of Chrdl1, synapses in the visual cortex do not undergo the physiological switch from calcium-permeable to GluA2-containing calcium-impermeable channels, and therefore they contain less GluA2 [106,107,108]. As a consequence, the removal of one eye after the closure of the critical period (monocular enucleation) in adult Chrdl1-deficient mice resulted in a significantly increased plasticity and remodeling in the binocular zone of mutant mice, with respect to wild-type. The loss of Chrdl1 results therefore in an extended critical period plasticity that remains open up to adulthood [109]. In humans, many different mutations in Chrdl1 that cause X-linked megalocornea1, a disease characterized by enlarged corneas and an increased risk of presenile cataracts, have been identified. Unexpectedly, these patients display a superior verbal IQ, verbal memory and executive skills [110].

### 5.2. Role of Astrocytes in Adult Forms of Synaptic Plasticity

Synaptic plasticity is the ability of a synapse to modify its structure in response to external stimuli, which persists through all the life and allows the continuous modifying of brain circuits in response to experience. Synapses can increase their strength (long term potentiation, LTP) or decrease their strength (long term depression, LTD) in response to alterations in neuronal activity. Synaptic scaling is a form of homeostatic plasticity with a compensatory up-regulation of synaptic activity in response to prolonged periods of activity deprivation. Both forms of plasticity play together in a concerted fashion, and astrocytes recently came out as possible candidates orchestrating these forms of plasticity; for further insights on this topic see [111].

In neurons, one of the major structural changes following LTP or LTD concerns the appearance or retraction of dendritic spines, and therefore the change of synapse number upon LTP or LTD induction. Similarly, astrocytic processes undergo rapid changes in volume and motility. Astrocyte PAPs are extremely sensitive to activity-driven changes at the synapse. The induction of LTP transiently enhances the motility and retraction of PAPs, allowing growth of the postsynaptic dendritic spine to occur. Subsequently, they will adjust their coverage to adapt to the new structure of the synapse [112,113].

The glial factor SPARC, already mentioned for its role in controlling synaptogenesis, is also fundamental to limit the number of synaptic AMPAR levels in the adult. SPARC KO mice in fact display an abnormal accumulation of surface AMPAR at the synapse, and enhanced excitatory glutamatergic neurotransmission, resulting in an altered NMDAR/AMPAR synaptic ratio and impaired LTP [20].

In addition, ephrinB1, which is expressed in adult mice, acts as a regulator of synaptogenesis in the adult hippocampus, and affects mouse learning behaviors. Of note, ephrinB1-deficient mice display immature dendritic spines, reduced evoked synaptic firing in CA1 hippocampal neurons and the proper acquisition of fear memory, but enhanced contextual fear memory recall. Moreover, astrocytic ephrin-B1 competes with the neuronal ephrin-B1 and triggers the astrocyte-mediated engulfment of ephrin B receptor-containing synapses via trans-endocytosis. This study demonstrated that astrocyte ephrinB1 is crucial to limiting new synapse formation in the adult hippocampus upon learning processes [114]. The same group, using an ephrin1B-overexpressing mouse model, recently investigated the mechanism through which astrocytic ephrin 1B influences the learning process. They demonstrated that alterations in contextual fear learning processes are characterized by the reduced formation of new spines, more so than impaired spine maturation [115]. Furthermore, astrocyte-derived ephrin 1B regulates synapse remodeling in the hippocampus in a mouse model of traumatic brain injury, through an increased expression of STAT3 in astrocytes [116].

Finally, new interesting roles in synaptic plasticity are emerging for lipid components that are produced and released by astrocytes, including astrocyte-secreted cholesterol. Astrocytic cholesterol synthesis is the primary source of cholesterol in the CNS. Its function is not confined to supporting synaptogenesis [117], but it has a fundamental role also in synaptic plasticity. The regulatory element binding proteins (SREBPs) control cholesterol production, ensuring an appropriate level in the brain. In the hippocampus, reduced cholesterol synthesis affects spine development and modification. In fact, in mice defective for SREBPs, both short-term and long-term plasticity are impaired [46,117].

Alterations in astrocyte lipid metabolism are associated with synaptic dysfunctions. For example, in Huntington disease, the mutant huntingtin protein in astrocytes decreases SREBP maturation, leading to impaired cholesterol biosynthesis and secretion, thus affecting synapse number and activity [118]. Cholesterol metabolism is significantly altered in Huntington’s disease patients. These alterations are associated with a striking reduction of the cholesterol 24-hydrolase (CYP46A1) expression in the putamen of patients. CYP46A1 is crucial for brain cholesterol elimination, because it mediates cholesterol hydroxylation. Of note, a gene therapy approach with CYP46A1 has been recently tested in a mouse model of Huntington’s disease [119].

In addition, in the autosomal recessive Niemann-Pick disease type C (NPC), which is characterized by an accumulation of unesterified cholesterol in late endosomes/lysosomes (LE/L), mutations in the NPC1 gene lead to impaired cholesterol transport in astrocytes. NPC affects neurological and psychiatric functions, including learning difficulties and cognitive impairments, as well as various internal organs. Affected neurons display higher cholesterol content in the soma and reduced cholesterol content in the distal axons. It is conceivable that some of the neurological deficits in NPC disease might be due to a deficiency of cholesterol in axons [120].

Apolipoprotein E4 (apoE4) is the main carrier of cholesterol, and it is the main genetic risk factor associated with sporadic Alzheimer’s disease (AD). It has a role in controlling Aβ formation through the regulation of lipid rafts functions in vivo [121]. Indeed, cholesterol is found to be enriched in the brain plasma membranes of AD patients. The cholesterol level increases throughout the course of clinical disease, and greater increase was observed when the disease progresses [53]. Because cholesterol is a multifunctional metabolite, abnormal cholesterol metabolism by ApoE4 would lead not only to altered cholesterol transport to other cell types, but also to functional deficits in astrocytes, as observed in neurons [122].

Regarding homeostatic plasticity, which allows neural circuits to elevate or reduce the activity of the full neuronal network in order to induce robust compensatory changes in the strength of the excitatory and inhibitory synapses. It maintains appropriate levels of excitability and connectivity despite changes in the surrounding environment brought about by metabolism and experience-dependent plasticity. Without homeostatic synaptic scaling, neural networks can become unstable and perform suboptimally [123,124]. Glial TNFα plays a fundamental role in this process by stimulating the rapid exocytosis of AMPARs calcium-permeable GluA2-lacking AMPARs, and the simultaneous endocytosis of GABA-A receptors, thus strongly affecting the balance of excitation-to-inhibition (E/I) balance [44,111,125,126,127,128]. In addition to regulating post-synaptic receptor content, glial TNF may also regulate pre-synaptic neurotransmitter release [125,129,130]. Consistently TNFα-deficient mice displayed normal LTP and LTD, but they did not undergo homeostatic synaptic scaling, demonstrating that TNFα is required for the upregulation of AMPARs upon depression [131].

## 6. Conclusions

Astrocytes are an essential “component” of the synapse, together with the pre- and postsynaptic compartments. They play important roles in synapse formation, maturation and elimination, as well as in the regulation of many aspects of synaptic activity, such as synaptic plasticity.

In this review, we presented an updated list of the astrocyte-secreted molecules fundamental for synaptogenesis. We focused on three main groups of molecules that act in concert to promote synaptogenesis, describing newly discovered mechanisms. Further, we went through some recent astrocyte-derived proteins and lipids implied in direct and indirect phagocytosis, which mediates synaptic refinement during the period of synaptic maturation. Lastly, we reviewed astrocytes’ contribution to the adult forms of synaptic plasticity.

The idea of “tripartite synapse” back in 1999 [132] proposed for the first time the contribution of astrocytes to synapse formation and function. Since then, many studies have discovered a number of molecules involved in this process, but this increasing knowledge still has some gaps. For example, do these molecules act in a manner widespread into the brain, or in a region-specific manner? Understanding in more detail the basis of astrocyte diversity would allow us to thoroughly define the specific role of each type. This would also contribute to dissecting which functions are redundant and which cannot be compensated.

Moreover, emerging evidence points to a role of astrocytes in human brain diseases, such as ASD and other neurodevelopmental diseases. This area of investigation is still in its infancy. In this respect, it will be important to investigate how astrocytes modulate synaptic development and function in the circuits that mediate cognition, emotion and social function. At present, this is a very ambitious goal that could ground unforeseen possibilities in the treatment of brain diseases.

## Figures and Tables

**Figure 1 cells-09-02173-f001:**
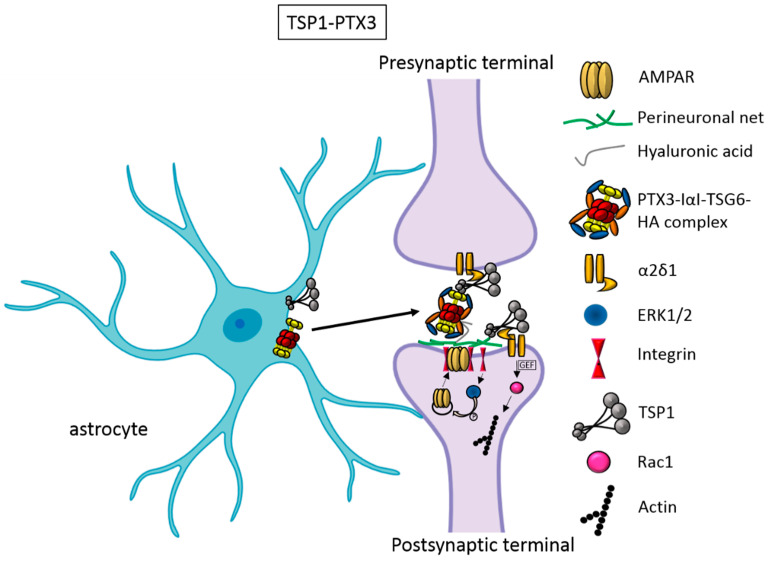
TSP1 and PTX3 cooperate to promote early formation of functional excitatory synapses. TSP1 binds to the α2δ1 receptor on the presynaptic terminals stimulating an increase in structurally normal but silent synapses, and to the α2δ1 receptor at the postsynaptic site, activating Rac1 and stimulating actin remodeling to promote spinogenesis. PTX3 promotes the functional maturation of these synapses by recruiting AMPA receptors at the synapse.

**Figure 2 cells-09-02173-f002:**
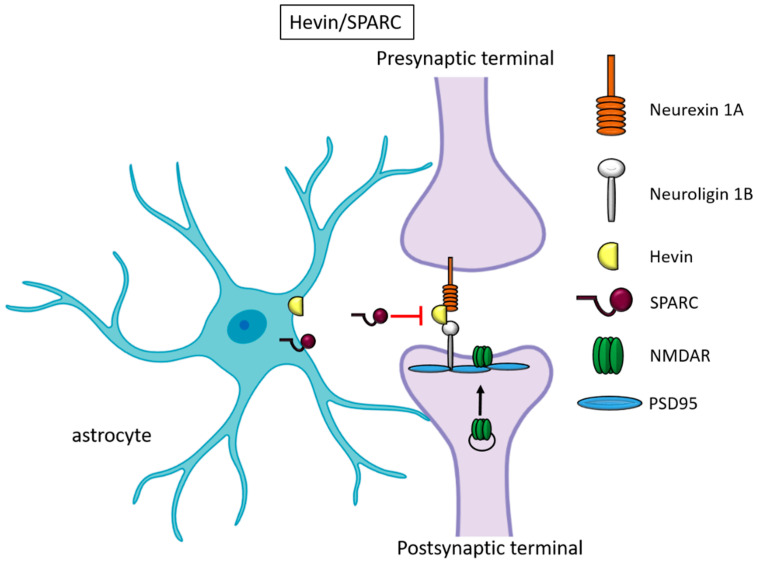
Hevin and SPARC. Astrocyte-secreted Hevin bridges presynaptic neurexin and postsynaptic neuroligin to favor the recruitment of PSD95 and NMDAR subunits at the synapse. SPARC on the contrary antagonizes Hevin’s effect with a still unknown mechanism.

**Figure 3 cells-09-02173-f003:**
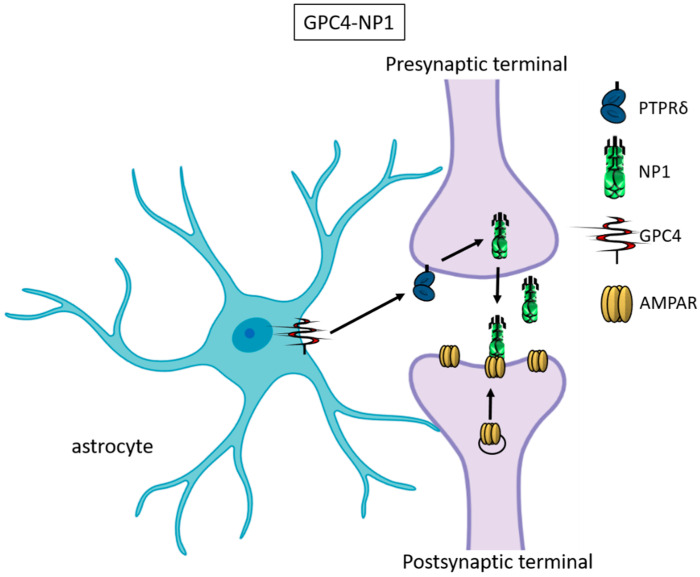
Glypican 4 and Neuronal Pentraxin 1. Astrocyte-derived Glypican 4 induces NP1 release from neurons through the PTPRδ receptor. NP1 stimulates AMPA receptors’ clustering on the postsynaptic terminal, making the synapse functional.

**Table 1 cells-09-02173-t001:** Main synaptogenetic astrocyte-secreted molecules.

	Name	Type of Synapse	Molecular Pathway	Effects on Synapses	References
**Synapse formation**					
	Thrombospondin 1 &2 (TSP 1&2)	excitatory	α2δ-1-> Rac	Promote silent synapses formation and actin remodeling at the spine	[14,15,16]
	Hevin	excitatory	presynaptic NRX1a and postsynaptic NL1B	Promotes silent synapses formation	[17,18,19]
	SPARC	excitatory	dominant-negative on hevin neuronal partners	Antagonizes hevin-induced synapses	[17,20]
	ApoE/cholesterol	excitatory	steroid and hedgehog pathways	Promotes excitatory synapses formation and increases presynaptic strength and release probability	[21,22]
	BDNF	excitatory	erbB	Promotes excitatory synapses formation	[23]
	Estrogen	excitatory	ER-α receptor	Promotes excitatory synapses formation	[24]
	γ-protocadherin	excitatory and inhibitory	astrocytes/neurons contact	Promotes excitatory synapses formation	[25]
	TGFβ	excitatory and inhibitory	NMDAR/serine D/CAMKII	Promotes excitatory and inhibitory synapses formation	[26,27,28]
**Synapse maturation**					
	PTX3	excitatory	β3 integrin/MAPK	Leads to functional activation of GluA-containing silent synapses	[29]
	Glypican 4&6	excitatory	RPTPδ/NP1/GluA1 recruitment	Induce functional synapses	[30,31]
	Chordin-like 1	excitatory	through CR repeats, but BMP independent	Induces maturation in GluA2-containing synapses	[32]
	Glial neuroligins (1&2)	excitatory and inhibitory	neuronal neurexins	Promote AMPA and NMDA receptors recruitment	[33,34]
	Wnt	NMJ	Repo (Drosophila)	Increases synaptic AMPAR	[35]
**Other**					
	Ephrin A3	excitatory	Rac	Promotes normal dendritic spine morphology	[36,37]
	Semaphorin 3A	excitatory and inhibitory	plexin A/neuropilin 1 receptor complexes	Positional cue required for proper establishment of motor neuron and sensory neuron circuit formation	[38]
	Neuregulin 1	excitatory and inhibitory	erbB	Guides tangential migration of cortical GABAergic interneurons and radial migration of differentiating pyramidal neurons	[39]
	BMP	excitatory and inhibitory	BMP receptor	Maintains the homeostasis of the synaptic microenvironment	[40]
	Maverick	NMJ	Gbb-dependent retrograde signaling (Drosophila)	Coordinates pre- and postsynaptic maturation	[41]

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
