# Peer review of "Astrocytic Factors Controlling Synaptogenesis: A Team Play"

_cells, 2020, doi:10.3390/cells9102173_

Round 1

Reviewer 1 Report

“Astrocytic factors controlling synaptogenesis: a team play”

The paper reviewed recent literature, showing correlations between astrocyte alterations and neurological disorders, particularly neurodevelopmental disorders.  This review is well-written, in that it gives a fundamental understanding of the known astrocytic factors and effects governing synaptogenesis.  Strengths of this paper are that it puts into perspective the possible areas for further research.  It clearly states the newer techniques, which can be used to further understand these pathways and effects. Suggestions are relatively minor and include

  • Line 23-25, pg. 1 – Consider rewriting this sentence as it took several times of reading to reach an understanding.

  • Figure 1 is very hard to see and introduces many concepts prior to their introduction in the text. Consider introducing each subpanel separately as figures with the corresponding subsection of text.

  • In Figure 1 text – “antagonizes” misspelled on line 15.

  • Line 21-25, pg. 4 – This is a very long sentence that contains a lot of information. Readers would likely benefit from this being 2-3 sentences.

  • Line 38-40, pg. 4 - Would suggest rewording of final paragraph sentence to clarify statement. The connection to the subsequent paragraph is unclear.

  • Line 55, pg. 5 - 1st mention of ASD should be spelled out.

  • Line 86, pg. 5 – change to Alzheimer’s.

  • Line 155-156, pg. 7 – Gender is a human construct. Sex is the correct term to be used, and it should be clarified that the astrocytes are from females (i.e., there are no “female astrocytes”).

  • Hevin is sometimes capitalized and sometimes not.

  • Line 234, pg. 8 –Missing “and” between channels and therefore.

  • Conclusion (section 6) reads as if it is disconnected from the rest of the paper. New animal models and syndromes are mentioned (i.e. Mecp-2null mice, Rett and Down syndromes). The body of the paper mentions several new techniques, by which to study astrocyte interactions, as well as specific astrocyte types in different areas of the brain.  As mentioned in the introduction, the paper covered the current data available regarding astrocyte’s roles via secreted molecules, regulation, and control of synapse number.  In compiling this data, the paper also states where pathways are not well known or unexpected results were found.  My recommendation would be that the conclusion redefine these areas for further investigation.  Examples of statements that would lead to the conclusion for further investigation include:  Lines 45-49, 58-60, 104-106, 128-130, 188-191, 215-217, 240-241, 248-250.

Reviewer 2 Report

The review by Fossati and colleagues is focused on astrocyte-derived molecules that regulate synaptic formation, degradation and function. Relevant literature is well discussed and presented by the authors and would be interesting for scientific community. I have only a few minor comments:

Table 1: please, consider making various sub-sections within the table that would correspond to sub-section of the text or to class of molecules

Fig 1 is not easy to read: please, increase font in all three boxes of the figure and also increase individual symbols that indicate signalling pathways.

Conclusions: please, briefly summarise the information in the review, providing overview for future research and impact this research will have – please, avoid discussing here new information/signalling pathways related to astrocyte role in synapse formation

Lines 30-32: please, rephrase the sentence

Lines 180-185: extremely long sentence, very difficult to follow

Reviewer 3 Report

The manuscript by Fossati et al. reviews important and recent findings showing the critical role played by astrocytes in synaptogenesis through a plethora of astrocytic-derived factors that are released during the different stages of brain development.

This paper is clearly written and covers the relevant literature in the field.

Minor comments:

1. the authors might quote and discuss the paper by Dowling & Allen (2018) showing that lower expression levels of GPC4 have been linked to autism spectrum disorder (ASD) and that mice lacking this heparan sulfate display behavioral alterations that are consistent with loss of GluA1.

2. The “Conclusions” paragraph, primarily focused on astrocytes and MeCP2 (RTT), should better recapitulate the whole matters discussed throughout the manuscript.
